# Reporter Phage-Based Detection of Bacterial Pathogens: Design Guidelines and Recent Developments

**DOI:** 10.3390/v12090944

**Published:** 2020-08-26

**Authors:** Susanne Meile, Samuel Kilcher, Martin J. Loessner, Matthew Dunne

**Affiliations:** Institute of Food Nutrition and Health, ETH Zurich, Schmelzbergstrasse 7, 8092 Zurich, Switzerland; susanne.meile@hest.ethz.ch (S.M.); samuel.kilcher@hest.ethz.ch (S.K.); martin.loessner@ethz.ch (M.J.L.)

**Keywords:** bacteriophage, reporter phage, genetic engineering, luciferase, CRISPR-Cas editing, bacterial detection

## Abstract

Fast and reliable detection of bacterial pathogens in clinical samples, contaminated food products, and water supplies can drastically improve clinical outcomes and reduce the socio-economic impact of disease. As natural predators of bacteria, bacteriophages (phages) have evolved to bind their hosts with unparalleled specificity and to rapidly deliver and replicate their viral genome. Not surprisingly, phages and phage-encoded proteins have been used to develop a vast repertoire of diagnostic assays, many of which outperform conventional culture-based and molecular detection methods. While intact phages or phage-encoded affinity proteins can be used to capture bacteria, most phage-inspired detection systems harness viral genome delivery and amplification: to this end, suitable phages are genetically reprogrammed to deliver heterologous reporter genes, whose activity is typically detected through enzymatic substrate conversion to indicate the presence of a viable host cell. Infection with such engineered reporter phages typically leads to a rapid burst of reporter protein production that enables highly sensitive detection. In this review, we highlight recent advances in infection-based detection methods, present guidelines for reporter phage construction, outline technical aspects of reporter phage engineering, and discuss some of the advantages and pitfalls of phage-based pathogen detection. Recent improvements in reporter phage construction and engineering further substantiate the potential of these highly evolved nanomachines as rapid and inexpensive detection systems to replace or complement traditional diagnostic approaches.

## 1. Introduction

The development of quick and reliable methods for pathogen detection and identification is critical to improve the prevention and treatment of bacterial diseases in various fields from food production to health care. While culture-based detection remains the gold standard for detection and identification of bacterial pathogens, it can be time and labor intensive, typically requiring more than 48 h to allow selective bacterial growth to ensure reliable detection [1,2]. Within the clinic, early microbial identification is important for ensuring patients receive optimal antibiotic treatment. Approximately 30–50% of patients receive ineffective antibiotic therapy because physicians must treat immediately with first-line, broad-spectrum antibiotics until the results of culture-based detection are available [3]. For example, blood culture assays require 48 to 72 h to complete [4], with fastidious organisms such as *Bacillus* species and HACEK (*Haemophilus* species, *Aggregatibacter* species, *Cardiobacterium hominis*, *Eikenella corrodens*, and *Kingella* species) organisms requiring several days to produce a conclusive result [5]. This not only affects patient survival due to the prescription of inappropriate antibiotics [6,7] but the misuse of antibiotics is a direct contributor to the global spread of antimicrobial resistant bacteria [8,9,10,11].

The increasing scale of production and global distribution of food goods, especially fresh fruit, vegetables and ready-to-eat products, makes quick and accurate microbial detection essential for ensuring circulation of high-quality and safe foods. Again, conventional culture-based methods are incommensurate with the quick turnaround time required by food producers today [12]. For example, ISO 11290-1:2017 guidelines for culture-based *Listeria monocytogenes* detection recommends a minimum of 24 h for colony formation on chromogenic agar, with an additional 24 h for slow-growing bacteria and an additional 96 h to enable complete morphological identification of colonies (e.g., phospholipase activity) [13]. As such, food products requiring test results prior to their release (positive release) are placed in temporary quarantine, which costs money and affects product half-life. Nevertheless, food products can still arrive on supermarket shelves without satisfactory microbial assessment, leading to costly product recalls as well as the spread of life-threatening foodborne disease outbreaks [14]. In Europe alone, over 23 million annual illnesses and 5000 deaths are associated with the consumption of contaminated food products [15]; in 2018, EU member states reported 5146 foodborne outbreaks, mostly from *Salmonella*-contaminated produce [16]. To meet the demands from the clinic and food producers for improved bacterial detection, a multitude of rapid diagnostic systems have been established that attempt to circumvent the need for extensive selective culturing.

While culture-based methods remain the mainstay diagnostic, clinical testing is becoming increasingly reliant on culture-independent diagnostic tests (CIDTs) such as nucleic acid amplification or ELISA-based antigen detection [17,18,19], as well as matrix-assisted laser desorption/ionization time-of-flight mass spectrometry (MALDI-TOF-MS) and whole-genome sequencing (WGS) for detection of bacteria [20,21]. The advantage of these approaches is the potential for automation, making them reproducible and easy to use while also allowing sensitive detection of non-culturable organisms and polymicrobial infections (multiplex detection). However, the specificity of these approaches can be affected by detection of closely related, non-target species generating false-positive results [22,23]. Furthermore, nucleic acid-based methods lack the ability to differentiate between DNA from viable and dead bacterial cells, meaning they can assess microbial viability retrospectively, i.e., changes in nucleic acid levels over a given timeframe; however, they are incapable of determining viability within discrete samples, making identification of only viable (and potentially infections) bacteria extremely challenging. The adoption of CIDTs, WGS and MALDI-TOF-MS by the food industry is also a significantly bigger challenge than clinical samples due to the lower levels of bacterial contaminants typically present and the greater variability in product matrices where heterogeneous levels of fats, proteins and debris can impede bacterial detection. A robust and sensitive alternative is to implement phage-based diagnostics.

Bacteriophages (phages) are environmentally ubiquitous viruses that infect specific bacterial hosts. The ability of phages to target and kill bacteria down to the species or even strain level, has led to the resurgence of interest over the last decade in their use as therapeutics (i.e., phage therapy) against antibiotic resistant bacterial infections [24,25,26]. Phages are also used as biocontrol agents to tackle foodborne bacteria such as *Listeria* and *Salmonella* [27]. Phages have evolved highly efficient mechanisms to attach to bacteria under various harsh environments that provide a clear advantage over other affinity-based system for bacterial recognition in complex sample matrices. The ability of phage particles to bind with high affinity and specificity to their target bacteria has led to their use as affinity molecules instead of other bio-probes such as antibodies in biosensor-based assays [28,29,30] and for bacterial enrichment and detection when conjugated to magnetic nanobeads [31] (Figure 1A). A drawback of using whole-phage particles exclusively as affinity molecules is their relatively large size and (unless they are inactivated) their basal lytic activity that may destroy target bacteria before downstream detection can be completed. The alternative to whole-phage bio-probes is to use phage proteins that confer host binding (Figure 1B,C). For example, cell wall-binding domains (CBDs) of phage endolysins have been successfully used for detecting various Gram-positive bacteria, e.g., *Listeria* [32,33], *Bacillus cereus* [34], and *Clostridium tyrobutyricum* [35]. Phages recognize their bacterial hosts using specialized receptor-binding proteins (RBPs), identified as tail fibers and tailspikes, which initiate the attachment of the phage to specific receptors on the bacterial cell wall [36,37]. Various RBP-based detection assays have been developed for detecting *Salmonella* [38,39], *Shigella* [40], and *Pseudomonas aeruginosa* [41]. By combining *Listeria*-specific CBDs and RBPs, a glycotyping system was also recently developed for discerning different *Listeria* serovars [42]. In addition to using the ability of phages or phage proteins to bind a distinct host range of bacteria, many research groups have developed highly efficient reporter phage systems that exploit the rapid infection, DNA replication, and phage production kinetics within infected target cells (Figure 1D–G). This can be achieved by simply monitoring an increase in phage titer (Section 2), or by quantifying the production of phage-encoded heterologous proteins such as fluorescent proteins, luciferases, or other enzymatic reporters (Section 5). This review focuses on the design principles that govern the engineering of heterologous reporters into phages (Section 3) and how these phages can be used to detect a wide variety of clinically and industrially important bacterial pathogens.

## 2. Phage Amplification-Based Detection

After successful host cell adsorption, a phage typically injects its genome and internal proteins into the bacterium and manipulates host metabolic processes to begin production of progeny phages, which are subsequently released upon host cell lysis. The number of progeny phages produced and the time required (latent period) can vary greatly between different phages and the hosts they encounter. For instance, coliphage T4 produces 150–200 progeny phage per infected cell within 25 min under optimal laboratory conditions [65]; however, under environmental and unfavorable conditions, phage amplification (i.e., burst size and latent period) can be affected [66]. Nevertheless, phage amplification can occur quickly within a contaminated sample to generate a large number of progeny particles. An increase in phage numbers indicates the presence of a susceptible host within the sample. The change in phage titer can be determined by traditional plaque assays using suitable indicator strains [52,67,68], by physical detection of the phage particles using ELISA-based assays [69], or as discussed below, by measuring the increase in phage nucleic acid content. Sensitivity of these assays can be improved by capturing and enriching the progeny phage particles [70] by using lateral flow assays [71,72,73,74] or (magnetic) bead-based enrichment. For example, antibody-conjugated beads were used to isolate amplified MS2 phages from solution for detection of *Escherichia coli* [75]. In addition to detecting whole-phage particles, real-time PCR methods have also been developed for detecting the increase in phage DNA generated during infection [55,68,76]. For instance, Griffiths and colleagues coupled capture of pathogenic *E. coli* and *Salmonella* Newport from food samples by phage coated paper dipsticks with qPCR-mediated detection of progeny phage DNA [55]. Phage RNA has also been used as a biomarker. For instance, surrogate marker loci in engineered mycobacteriophages [77,78] or simply the phage’s natural RNA [54] have both been detected using qPCR methods. Methods that are based on reverse transcription of phage RNA can offer enhanced sensitivity and are less prone to false-positive results than DNA amplification-based detection systems [79].

In addition to progeny phage detection, cell lysis releases an abundance of cellular content that can be used as diagnostic markers. For instance, ATP is easily detectable after release from cells using a bioluminescence reaction with firefly luciferase [80]. This kind of bioluminescence-based sensing of ATP released from cells after infection with phage K enabled detection of *Stapylococcus* spp. in fluid from orthopedic artificial joints [53]. Other methods have visualized the change in redox reactions by *Salmonella enterica* Typhi and Paratyphi after phage infection [81], or measured the release of eDNA from phage infected *E. coli* [82]. Table 1 provides a detailed overview of phage amplification-based detection assays for bacterial pathogens. A downside of most phage amplification-based assays is the requirement for completion of the phage lytic cycle to allow the release of biomarkers or progeny phages for detection. In contrast, many reporter phage-based systems do not require cell lysis or completion of the phage infection cycle to detect infected cells. Host adsorption and phage genome injection followed by reporter gene expression is generally sufficient to produce a detectable signal, allowing reporter phages to circumvent the gamut of intracellular phage defense mechanisms that can inhibit various stages of the phage infection cycle [83,84].

## 3. Design Rules for Reporter Phage Engineering

Compared to measuring phage amplification or released cell content, reporter proteins can be overexpressed within the bacterial host using strong promoters to produce an intense and amplified signal for instant detection upon substrate addition, providing a more sensitive diagnostic assay than relying on phage lysis alone. As detailed in Table 2, an abundance of reporter phage systems have been developed in the last four years using different engineering platforms and reporter genes. As important as the choice of engineering technique seems to be, the question of how to design the reporter phage is of paramount importance. Four principles for engineering an effective reporter phage are:The phage has to be able to infect or transduce the reporter gene into the target bacteria. Therefore, phages capable of infecting a broad spectrum of representative strains for the target pathogen should be used.Both the reporter gene and the method of detection have to be suitable for the target bacteria, expected microbial content, and the sampling environment. Inclusion of a pre-enrichment step is highly recommended when low bacterial numbers are expected in the sample, for instance, the zero tolerance policy for common food pathogens (e.g., *Salmonella, Campylobacter* and *E. coli*) necessitates a pre-enrichment of food samples to ensure bacterial amplification to a minimal threshold for detection. In addition, as food components can affect fluorescence-based detection or physically block biosensors, detection from complex matrices should be coupled with an initial capture and enrichment step, e.g., bead-based magnetic separation.Identify a suitable region within the phage genome that allows for integration of a heterologous reporter gene without disrupting infectivity. Furthermore, lysogenic phages can be modified to become strictly lytic by targeted deletion of genes involved in prophage integration and maintenance. The removal of such genes makes room for additional payloads within space-limited phage genomes. For example, Kim et al. 2017 deleted whole regions unrelated to phage infectivity, such as the integrase (*int*), adenine methylase (*am*), *O*-acetyltransferase (*oac*), and anti-immunity protein (*aip*) genes and replaced them with the *luxCDABE* operon [88]. In *Listeria* phages A500 and A006, the lysogeny control module was removed, rendering the phages strictly lytic (Meile et al. 2020). DS6A-derived mycobacteriophages Φ^2^GFP12 and Φ^2^GFP13 were created by replacing two independent regions including the integrase gene or *mazG* with the L5 promoter driven mVenus reporter cassette [58]. In the *E. coli* phages IP008BK and IP052BK, the non-essential small outer capsid gene *soc* was replaced with the cytochrome *c* peroxidase gene *ccp* [89].Reporter gene expression should be tuned to ensure sufficient production and subsequent signal generation for detection. For instance, a heterologous strong promoter can be used or the reporter gene can be inserted into a region controlled by a strong endogenous promoter (e.g., structural genes). For instance, expression of NanoLuc from an early promoter controlling the anti-CRISPR locus of *Listeria* phage A006 provided rapid protein production and early detection during the latent period [90]. Still, many reporter genes are inserted downstream of the strongly expressed but rather late endogenous promoters of the major capsid gene [62,91] the endolysin gene [62], or the tail spike protein [88]. Some examples of recently used constitutive exogenous promoters are P_L(L5)_ for mycobacteria [58,92,93,94], and PrplU [95] and phi10 T7 [96] for *E. coli.* In addition, the *N*-terminal leader sequence pelB was added to increase soluble heterologous gene expression and direct translated reporter proteins to the periplasmic space ([97,98,99]).

In addition, various reporter proteins do not require cell lysis or completion of the phage infection cycle to produce a signal. Genome injection and potentially some degree of DNA replication is therefore sufficient [58,62]. In fact, an intact phage genome may not even be required: non-replicative phage particles carrying a luciferase-containing plasmid (fittingly named Smarticles™ (Roche/Geneweave, Los Gatos, CA, USA)) simply transduce the luciferase gene into the target bacteria for detection [100]; however, this approach does remove the possibility of signal amplification from secondary phage infections.

## 4. Engineering Platforms Available for Reporter Phage Generation

Current approaches for engineering of phages can be assigned to three categories: (i) direct cloning; (ii) homologous recombination with or without CRISPR-Cas counter selection; and (iii) whole genome activation, a.k.a. “rebooting”.

### 4.1. Direct Cloning

Phages can be cloned using phage vectors (phasmids or phagemids), which are plasmids with an additional origin of replication and packaging sequence originating from a phage. This approach is quick and efficient due to the ability to insert genes using standard cloning procedures (e.g., restriction enzyme digestions and insertion into multiple cloning sites) and phasmids can be simply propagated as plasmids or lytically as phages. However, the application is currently limited to mycobacteriophages [116,117] and Gram-negative targeting phages, such as lambda, M13 and T7 [118,119,120]. Recently published fluoromycobateriophages described in Table 2 were engineered using phasmids following the same strategy: TM4 shuttle phasmids are cloned in vitro to contain the desired reporter gene (e.g., GFP or RFP), and the phasmid DNA is then packaged into phage λ particles for amplification in an intermediate *E. coli* host before transformation into a fast growing *M. smegmatis* strain mc^2^155 to produce the final infectious virions [56,57] (Figure 2A). A major limitation of using this system is the relatively small packaging capacity of the lambda phage capsid (~53 kb), meaning larger genomes are unsuitable for phasmid construction, especially for simultaneous insertion of reporter genes [121].

An alternative direct cloning platform used by Nugen and colleagues is the T7Select^®^ system (Novagen, Gauteng, South Africa), originally designed as a phage-display vector, which is capable of displaying peptides and large protein antigens (up to 1200 amino acids) on the capsid [122]. By incorporating a stop codon downstream of the reporter cassette, heterologous reporter proteins are released instead of being displayed [96] (Figure 2B). This system was used to create various T7-based systems for detecting *E. coli* in water and food [60,97,98,105,106]. Not only is a commercial kit available for T7 phage engineering (i.e., T7Select^®^ (Novagen)), but as a model phage it has a genome suitable for insertion of heterologous genes and a well-characterized host range, making it an ideal phage for reporter phage-based assay development as can be observed by its frequent use over the last four years (Table 2).

### 4.2. Homologous Recombination Combined with CRISPR-Cas Selection

Phage engineering strategies based on homologous recombination make use of editing plasmids carrying the desired genome modifications flanked by homology regions. When host cells harboring editing plasmids are infected with wild-type phage, double homologous recombination can occur during replication of the phage genome, leading to the production of wild-type and recombinant phages (Figure 2C). Due to low recombination rates (10^−10^–10^−4^), screening can be extremely laborious and typically requires a suitable selection marker, such as antibiotic resistance in lysogens, a change in host range [118], or plaque morphology [91]. As an example, the temperate, *Bacillus anthracis* phage Wβ∷*luxAB*-2 was generated using spectinomycin resistance transduction as the selection marker. The phage carried *Vibrio harveyi luxAB* and the spectinomycin resistance (*spc*) cassette under the control of an optimized promoter (LuxAB-2) in a non-essential region of the genome that generated bioluminescence and provided spectinomycin resistance to the lysogen [101,123]. In the absence of selection markers, low recombination frequencies can be increased using the lambda red homologous recombination-mediated genetic engineering (recombineering) system [124,125,126], which is commonly used for modifying *E. coli* and *Salmonella* targeting phages such as the reporter phage ΦV10 for detecting Shiga toxin producing *E. coli* O157:H7 [102]. The system originates from phage lambda but can be expressed from the bacterial chromosome or a separate plasmid, making it widely applicable for engineering other phages (reviewed in detail by [127]).

Homologous recombination-based methods can be coupled with CRISPR-Cas systems to facilitate enrichment of recombinant phages by sequence-specific counter selection of wild-type phage genomes (Figure 2C). [128,129,130]. This approach has been developed for engineering Gram-negative and Gram-positive targeting phages. For instance, Jackson et al. 2016 employed the *E. coli* type I-E CRISPR-Cas system [128,131] to create a T7 based reporter phage carrying an alkaline phosphatase for colorimetric detection of *E. coli* [107]. Similarly, a *Listeria ivanovii* type II-A CRISPR-Cas system [132] was used for the modification of lytic *Listeria* phage A511 to create two variants of A511::*nluc* that transduce bioluminescence into *Listeria* spp. for detection [62]. While the use of CRISPR-Cas counter selection can greatly improve recombinant phage identification, a current bottleneck is the lack of well-characterized and programmable CRISPR-Cas systems that can be used for reporter phage engineering in different bacterial hosts.

### 4.3. Synthetic Genome Rebooting

Synthetic genome rebooting involves assembling a complete genome from individual PCR fragments (or synthesized DNA) for subsequent transformation into a surrogate bacterial host for phage production. One approach is to transform the genome fragments together with a yeast artificial chromosome (YAC) into *Saccaromyces cervisiae* featuring an efficient gap repair system to produce a full phage genome cloned into a replicative yeast plasmid [133,134] (Figure 2D). The assembled YAC–phage DNA is extracted and subsequently transformed into a bacterial host for virion production. By switching the tail fiber genes between synthetic T7-like phage genomes, this method could redirect phage host range between *E. coli, Klebsiella* and *Yersinia* hosts [134]. Recently, this approach was used to engineer *E. coli* phage λ-mKate, a lysogenic phage featuring a red fluorescent reporter to observe prophage induction in bacterial host cells within phagocytes [135]. A major restriction to this approach is the requirement to transform a bacterial production strain with the assembled genome phages. While this is possible for some Gram-negative bacteria, it can be challenging to transfer such large viral genomes across the thick wall of Gram-positive cells. To overcome this limitation for Gram-positive targeting phage engineering, a phage engineering method using a strain of L-form *Listeria* for rebooting of synthetic phage genomes was recently developed [136] (Figure 2E). L-forms are wall-deficient bacteria that retain metabolic activity and the ability of cell division [137,138,139]. Induction of L-forms occurs by prolonged subcultivation under selective pressure of cell wall targeting antibiotics in a medium that provides osmoprotective conditions, preventing the wall-deficient cells from hypotonic lysis. Using L-forms as surrogate hosts for the reactivation of synthetic, in vitro assembled genomes provides many advantages: multiple modifications can be achieved in a single step and no cloning of potentially toxic phage genes is required. Low-efficiency homologous recombination and subsequent screening can be avoided because only correctly assembled genomes and viable recombinant phages are isolated using this technology. While native phage genomes of different sizes and structures can easily be rebooted in L-forms [140], the current limitation of this approach is the assembly of large (>100 kb) genomes in vitro, using approaches such as Gibson assembly of multiple genetic fragments [141].

## 5. Overview of Reporter Phage Systems

### 5.1. Bioluminescence-Based Detection

The phenomenon of bioluminescence is found across a wide diversity of life including bacteria, fungi, insects and a variety of marine organisms [142]. Light (photon) emission occurs when a substrate is oxidized by a member of a class of enzymes called luciferases. Due to highly sensitive signal detection and ease of use, luciferase reporter phages have found broad application for bacterial detection. The ideal luciferase leads to bright and sustained light emission with low background. Preferably, the luciferase should be structurally stable in different environmental conditions. We recently compared the performance of several isogenic reporter phages encoding for different luciferases. Bacterial, cnidarian and crustacean luciferase coding sequences derived from *Vibrio harveyi* (*luxAB*), *Gaussia princeps* (*gluc*), *Renilla reniformis* (*rluc*) and *Oplophorus gracilirostris* (*nluc*) were inserted into the *Listeria* phage A500. The light-emitting properties of the NLuc reporter phage (A500::*nluc* ΔLCR) were clearly superior, indicated by a 100-fold larger increase in luminescence values compared to other reporters. Overall, the assay using the NLuc luciferase was highly sensitive and able to directly detect as few as three *L. monocytogenes* cells [62]. Nluc is an engineered luciferase (19 kDa) that produces a glow-type bioluminescent signal upon addition of its substrate (furimazine; signal half-life > 2 h) [143] and is widely used in reporter phage assays as detailed in Table 2. The first published reporter phage encoding NLuc is *E. coli* phage ΦV10 for detecting *E. coli* O157:H7 [102]. Another recent example is the broad host-range, *nluc*-containing Myovirus A511 (A511::*nluc*_CPS_) that detects a single *L. monocytogenes* cell in 25 g of various artificially contaminated food samples within less than 24 h. In addition to A511::*nluc*_CPS_-mediated detection, other *nluc Listeria* phages can be used for serovar differentiation of food isolates [62]. Furthermore, Dow et al. 2018 used acoustic separation and microfluidics to separate bacteria from blood cells and then employed the NLuc-reporter phage K1E for detection of *E. coli* [103]. A set of T7-based phages encoding a NLuc-carbohydrate-binding module fusion protein (NLuc-CBM) were evaluated for detection of *E. coli* in water and food samples [60,98,104,105]. One specific assay used cellulose-coated beads to concentrate and purify NLuc-CBM after its production from infected *E. coli* cells, enabling more sensitive detection down to 1 CFU/100 mL drinking water within 10 h [60].

The *luxCDABE* operon encodes for the luciferase (LuxA and LuxB) and the enzymes that produce its substrate (Lux C, D and E). Typically, only *luxA* and *luxB* are inserted into reporter phages and its substrate, a fatty aldehyde, is applied exogenously to the reaction solution [59]. Introducing a complete *luxCDABE* operon creates a substrate-independent reporter system; however, the relatively large size (approximately 6 kb) of the operon complicates phage engineering. Nevertheless, temperate *E. coli* phages HK620 and HK97 that use different packaging systems both proved functional upon integration of the complete *luxCDABE* operon and were used to detect *E. coli* in solution [95]. Similarly, ΦV10lux was engineered to contain the full operon for detection of enterohemorrhagic *E. coli* O157:H7 in food. In order to make room for the *luxCDABE* operon on the genome of this phage, non-essential regions were removed, which additionally converted the phage to a strictly lytic lifestyle. The operon itself was integrated behind the tailspike gene and expression driven from the endogenous phage promoter [88].

### 5.2. Colorimetry-Based Detection

Colorimetric signals arise from the result of enzymatic substrate conversion that can be visually interpreted. Beta-galactosidase (β-gal) is a glycoside hydrolase (*lacZ*) encoded in the *lac* operon of *E. coli*. T7 phages have been engineered to carry the *lacZ* gene, leading to β-gal expression during phage infection [63,64,87]. Upon release from the cell, the enzyme hydrolyses a colorimetric substrate for visual detection. For example, phage T7_LacZ_, in combination with chlorophenol red-β-d-galactopyranoside (CPRG) as a substrate for β-gal, was employed for antibiotic resistance profiling [63] and for detecting *E. coli* in food samples [64]. Similarly, T7 phages encoding alkaline phosphatase enabled detection of *E. coli* by hydrolysis of the substrate *p*-nitrophenyl phosphate (*p*NPP) to *p*-nitrophenol (*p*NP) [97,107] or upon reaction with nitro-blue tetrazolium chloride NBT and 5-bromo-4-chloro-3′-indolyphosphate *p*-toluidine salt (BCIP) [60,106]. The alkaline phosphatase could be functionalized with a cellulose-specific carbohydrate-binding module (CBM) from *Cellulomonas fimi* and consequently captured on magnetic cellulose [97] or cellulose filters [60] for detection. The latter enabled detection of 1 CFU/100 mL of *E. coli* in drinking water. Furthermore, detection of enterohemorrhagic *E. coli* in fresh produce was recently achieved using the cytochrome *c* peroxidase gene *ccp* as a reporter in recombinant phages PP01ccp [108], IP008BK and IP052BK [89] Bacterial detection using this system is based on the oxidation of cytochrome *c* and associated shift in absorbance at 550 nm after release of the reporter enzyme from the cells. Overall, colorimetric assays are straightforward and cost effective, but are limited greatly by the composition of the sample matrix, for instance, colored solutions can interfere with the visual readout and the pH can have a significant effect on substrate conversion [63]. Capture and detection approaches are therefore advisable when using colorimetric reporters.

### 5.3. Electrochemistry-Based Detection

Electrochemical biosensors are relatively simple and cost-effective while remaining sensitive and specific. Usually, electrochemical measurements are based on the detection of electroactive species and thus are not influenced by turbidity or color of the samples. Modulation of electrical properties is a result of redox reactions occurring among analytes. In addition to the use of phages as physical bio-probes in biosensors (reviewed in [144]), reporter proteins produced during phage infection can be used as the analyte for detection. T7-based reporter phages featuring the *LacZ* operon produced β-galactosidase (β-gal) that is released upon cell lysis. Enzymatic activity is detected by measuring the level of 4-aminophenol (PAP) produced upon hydrolysis of the substrate 4-aminophenyl-β-d-galactopyranoside (PAPG) by β-gal. The electroactive PAP product is subsequently monitored by amperometry (detection of ions in solution) [109]. Immobilization of biomarkers on the surface of electrodes can also serve to increase the sensitivity of electrochemical biosensors. Wang et al. engineered a T7-based reporter phage featuring a gold-binding peptide fused to an alkaline phosphatase (GBPs-ALP) that is released during cell lysis and binds directly to the gold biosensor surface. The activity of GBPs-ALP-coated electrodes was subsequently measured electrochemically using linear sweep voltammetry (LSV), which enabled detection of 10^5^ CFU/mL in drinking water after 2 h [110].

### 5.4. Fluorescence-Based Detection

The majority of fluorescence-based reporter phage assays developed in the last four years involve the detection of *Mycobacterium* and drug susceptibility testing (DST). Engineered fluoromycobacteriophages are combined with the drug of interest and the clinical isolates. Phage-mediated fluorescence only occurs in drug-resistant bacteria, which is detected by fluorescence microscopy or flow cytometry [94,112]. Due to its broad host range against several *Mycobacterium* species, the majority of recently employed fluoromycobacteriophages are derivatives of the temperate phage TM4 [145,146]. Additionally, these fluoromycobacteriophages are thermosensitive, thus they do not lyse their host cells at 37 °C. Thermosensitivity of the fluorophages ensures the survival and thus detection of drug-resistant hosts during DST. Recently, the TM4-derived phage Φ^2^GFP10 was used to detect low-frequency drug-resistant subpopulations of *M. tuberculosis* in vitro and in sputum from a South African TB patient [94] a second-generation fluoromycobacteriophage with optimized expression of a mCherrybomb gene in mycobacteria with improved fluorescent signal allowed shorter time to detection of *M. tuberculosis* [112]. mCherrybomb-Φ proved useful in a microscopy-based approach for detection of *Mycobacterium* spp. and determination of rifampicin resistance directly from Brazilian TB patient sputum within days. This phage was further used for evaluation of phage based DST and discrimination between *M. tuberculosis* complex (MTBC) and non-tuberculous mycobacteria (NTM) strains [111]. Other fluorescence-based assays for the detection of *E. coli* relied on the detection of the reporter with antibodies [106,115] or have been using a fluorescent substrate for the alkaline phosphatase reporter [114].

## 6. Future Perspective

Reporter phage-based assays combine host-specific binding with rapid intracellular phage multiplication and gene expression to provide highly sensitive detection. Nevertheless, the major limitations to practical reporter phage application are food matrix effects and the restricted host ranges of many phages that is intrinsically linked to phage resistance. The first limitation can be circumvented by coupling detection to specific capture of the target bacterium or the amplified reporter protein. Restricted host range and phage resistance are mechanistically interconnected and caused either by a lack of phage adsorption/receptor engagement or by the presence of intracellular defense mechanisms within individual strains. In therapeutic settings, this limitation is often circumvented by using multiple phages with complementary host ranges (the “phage cocktail”). With faster engineering protocols at hand, the cocktail approach could also be adopted in future reporter phage studies. In addition, synthetic biology and phage genome engineering may offer more targeted strategies to tune a limited number of well-characterized phage backbones towards the specific needs of each reporter phage application. This could be achieved by reprogramming the phage binding range through targeted RBP or baseplate engineering. Several recent studies suggest that redirecting and tuning host specificity is a viable option, at least for *Sipho*- and *Podoviridae* [134,147,148]. Counteracting intracellular defenses through phage engineering could be another viable strategy to expand the detection range of reporter phages. For example, RM systems can be counteracted through specific methylation within the phage production strains or CRISPR-Cas systems could potentially be inactivated via phage-mediated delivery of anti-CRISPR proteins. Due to the large variety and complexity of intracellular defenses, this strategy may be even more challenging and requires a detailed understanding of the underlying mechanisms. Despite these limitations, many reporter phage candidates already perform exceptionally well and will continue to offer an inexpensive and rapid alternative to culture-based and molecular diagnostics, which will be further improved through genetic engineering.

## Figures and Tables

**Figure 1 viruses-12-00944-f001:**
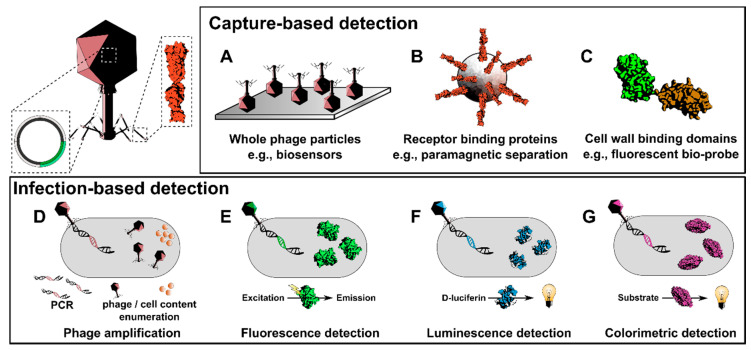
Overview of phage-based pathogen detection. *Capture-based detection:* The limited host ranges of phages towards a given genus, species or subspecies of bacteria make them ideal candidates for development into detection bio-probes. (**A**) The high binding affinity of whole-phage particles has led to their use as bio-probes in biosensors [28,29,30], or by conjugation with radioactive tracers [43] fluorophores [44], magnetic nanoparticles [31,45] or a combination of both [46] to label and enrich bacteria for detection. (**B**) Host specificity of phages is mediated by specialized receptor-binding proteins (RBPs) that provide equivalent binding capabilities as whole phages, but at a fraction of the size. Recently, RBPs have been applied in biosensors [47], ELISA-based assays [38,40], and for glycotyping *Salmonella* [39] and *Listeria* [32]. (**C**) Alternatively, cell wall-binding domains (CBDs) of phage endolysins have proven highly effective at detecting Gram-positive pathogens. CBDs have recently been used to detect *B. cereus* using biosensors [34] lateral flow assays [48] and magnetic enrichment-based detection [49]. In addition, *Listeria-*targeting CBDs were used to glycotype and identify *Listeria* serovars [42] and *Clostridium tyrobutyricum-*targeting CBDs have been employed for spore detection during cheese spoilage [35,50]. *Infection-based detection:* Infection of a bacterial host by lytic phage leads to rapid progeny phage production and ensuing cell lysis. (**D**) Released progeny phages [51,52] or bacterial cell content (e.g., ATP, DNA, RNA and bacterial proteins) provide excellent markers for downstream detection of the original bacterial host [53,54,55]. Alternatively, genetically engineered phages encoding reporters such as fluorescent proteins (**E**) [56,57,58], luciferases (**F**) [59,60,61,62] or hydrolyzing enzymes (e.g., β-galactosidase) (**G**) [63,64] are used. These phages express the reporter proteins during host infection to produce an amplifying fluorescent or bioluminescent signal upon the addition of a substrate. The rapid and sensitive nature of reporter phage-based systems has made them ideal tools for detecting low levels of viable, contaminating bacteria in many matrices, including foods, water and clinical samples.

**Figure 2 viruses-12-00944-f002:**
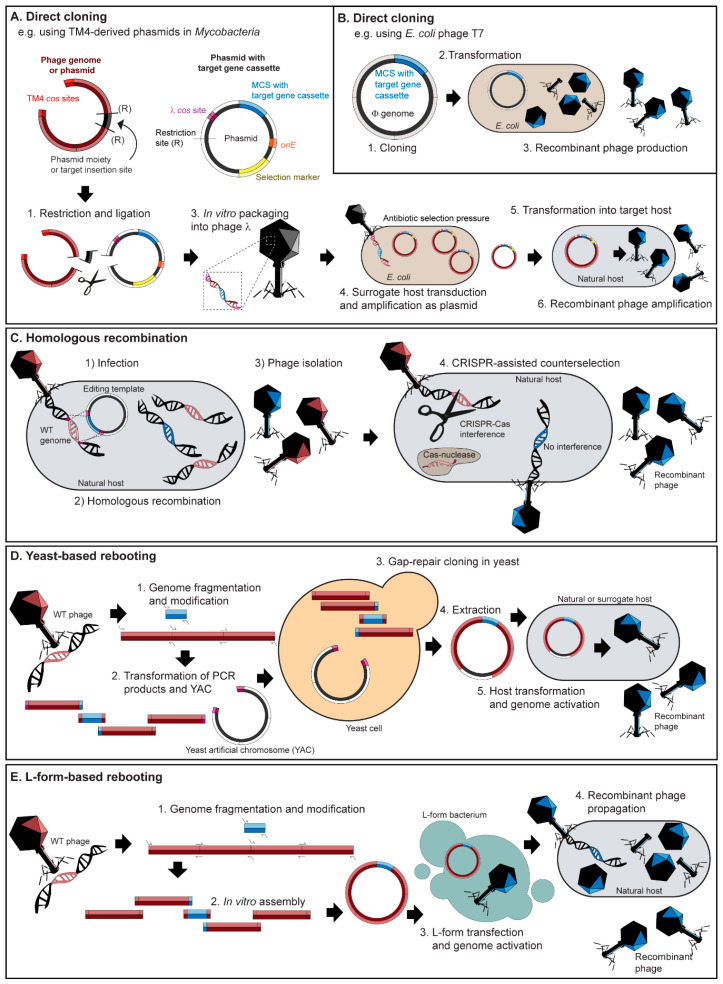
Overview of engineering strategies for reporter phage generation. (**A**) Heterogeneous reporter genes can be inserted into dual function shuttle phasmids capable of plasmid replication in *E. coli* and phage replication in a target host, e.g., *Mycobacterium* spp. (**B**) Reporter genes can also be directly inserted into phage genomes, e.g., T7Select^®^(Novagen). (**C**) Homologous recombination (and recombineering with bacteriophage-encoded homologous recombination systems, such as the coliphage λ Red system) involves the swapping of a reporter gene containing homology arms featured on an editing plasmid with the homologous region located within a phage genome. Homologous recombination can be combined with CRISPR-Cas counter selection to facilitate the removal of wild-type phages to improve identification of recombinant phages. (**D**) PCR products of a phage genome and a reporter gene insert are transformed alongside a linearized yeast replicon fragment from a yeast artificial chromosome (YAC) into yeast cells. The phage genome is assembled in the YAC vector by gap-repair cloning. YAC–phage DNA is extracted and directly transformed into the host bacterium leading to the production of recombinant phages. (**E**) A synthetic genome can be in vitro assembled (e.g., using Gibson assembly) from PCR fragments featuring a reporter gene insert. The synthetic genome can be transformed into L-form bacteria leading to genome activation and the release of viable phage for subsequent propagation on the phage host.

**Table 1 viruses-12-00944-t001:** Phage amplification-based detection assays published since 2016.

Phage	Target	Readout	Application, Sample	Application, Assay	DETL *	DETT *	Reference
Phage K	*Staphylococcus* spp.	bioluminescence	prosthetic joint sonicate fluid (SF)	detection within SF of infected prosthetic joints using an ATP bioluminescence assay	10^3^ CFU/mL	4 h	[53]
T7	*Escherichia coli*	fluorescence microscopy	food, various	visualization of eDNA release after phage induced lysis	10 CFU/mL	8 h	[82]
MS2	*Escherichia coli*	immunoassay	laboratory	phage amplification coupled, bead-based sandwich-type immunoassay	10^2^ CFU	3 h	[75]
A511	*Listeria monocytogenes*	immunoassay, SERS-LFI	laboratory	antibody-conjugated SERS nanoparticles as quantifiable reporter	5 × 10^4^ CFU/mL	8 h	[73]
Phage K	*Staphylococcus aureus*	MALDI-MS	laboratory	detection of *Staphylococcus aureus* and antibiotic susceptibility testing	n/a	n/a	[85]
Phage 10	*S.* Typhimurium	optical, absorbance	animal rectal swabs	absence of bacterial growth as indicator for phage activity	10^3^ CFU/mL	30 h	[86]
DN1, UP2, UP5	*S.* Typhi and Paratyphi	optical, colorimetric	laboratory	absence or delay of color change as indicator for phage activity, differentiation between serovars	10 CFU/mL	6 h	[81]
T7	*E. coli*	colorimetry	laboratory	detection based on the enzyme-induced silver deposition on gold nanorods detected by LSPR	10^4^ CFU/mL	n/a	[87]
ST560Ø	*Salmonella typhi*	plaques	laboratory, water	detection of viable but non-culturable (VBNC) state *Salmonella typhi* after starvation	n/a	n/a	[52]
PA phage	*Pseudomonas aeruginosa*	plaques	laboratory, water	detection of VBNC state *P. aeruginosa* after water disinfection by photocatalytic treatment	n/a	n/a	[67]
D29	*Mycobacterium avium* subsp. *paratuberculosis* (MAP)	plaques, PCR	whole blood	bacteriophage amplification-based detection from PBMCs, end-point PCR	n/a	n/a	[51]
D29	*Mycobacterium bovis*	DNA amplification	blood	phage amplification coupled RPA	10 CFU/mL	48 h	[68]
rV5, AG2A	*Escherichia coli*	qRT-PCR	food, various	capture by paper dipstick and PMMD of *E. coli* and *Salmonella* in spinach and ground beef	10 CFU/mL	8 h	[55]
CGG4-1	*Salmonella* Newport	qRT-PCR	food, chicken broth	paper dipstick-mediated capture and PMMD of *E. coli* and *Salmonella*	50 CFU/mL	8 h	[55]
Tb, Fz, Wb, S708, Bk	*Brucella abortus*	qRT-PCR	laboratory, simulated blood	PMMD of *Brucella abortus*	1 CFU/mL	72 h	[76]
Phage K	*Staphylococcus aureus*	qRT-PCR (RNA)	laboratory	PMMD and antibiotic susceptibility testing of *Staphylococcus aureus*	10^2^ CFU	3–5 h	[54]
Phage Gamma	*Bacillus antracis*	qRT-PCR (RNA)	laboratory	PMMD and antibiotic susceptibility testing of *Bacillus anthracis*	n/a	n/a	[54]

n/a, not available; DETL, detection limit, DETT; overall detection time; PMMD, phage-mediated molecular detection; RPA, recombinase polymerase amplification; * DETL and DETT are reported for individual assays as they were derived or provided from the source articles.

**Table 2 viruses-12-00944-t002:** Reporter phage-based detection assays published since 2016.

Phage	Target	Readout	Reporter	Application, Sample	Application, Assay	DETL *	DETT *	Lifestyle	Cloning Strategy	Reference
phiV10*lux*	*E. coli*	bioluminescence	LuxA, LuxB,	food, various	detection of *E. coli* O157:H7	13 CFU/ml	6 h	virulent (engineered)	homologous recombination (HR)	[88]
HK620	*E. coli*	bioluminescence	LuxA, LuxB,	environmental, water	detection in tap water	10^4^ CFU/mL	1.5 h	temperate	recombineering	[95]
HK97	*E. coli*	bioluminescence	LuxA, LuxB,	laboratory	detection of *E. coli*	n/d	n/d	temperate	recombineering	[95]
Wβ∷luxAB-2	*Bacillus anthracis*	bioluminescence	LuxA, LuxB, (*spc*^R^)	environmental, soil	detection of *B. anthracis* spores in soil	10^4^ CFU/g	6 h	temperate	HR	[101]
Wβ∷luxAB-2	*B. anthracis*	bioluminescence	LuxA, LuxB, (*spc*^R^)	environmental, water	detection of *B. anthracis* spores in pond, lake and brackish water	10–100 CFU/mL	8–12 h	temperate	HR	[61]
Y2	*Erwinia amylovora*	bioluminescence	LuxAB	environmental, plant material	detection and biocontrol of *E. amylovora*	4 × 10^3^ CFU/mL	1 h	virulent	HR	[91]
A511::*nluc*	*Listeria* spp.	bioluminescence	Nluc	food, various	detection of *Listeria* in food	1 CFU/25 g	24 h	virulent	HR + CRISPR-Cas counter selection	[62]
A006::*nluc* ΔLCR	*Listeria* spp.	bioluminescence	Nluc	laboratory	detection and serovar differentiation	1 CFU/mL	3 h	virulent (engineered)	L-form-assisted reactivation	[62]
A500::*luc* ΔLCR	*Listeria* spp.	bioluminescence	Nluc, RLuc, GLuc, LuxAB	laboratory	detection and serovar differentiation of *L. monocytogenes*	2–100 CFU/mL	3 h	virulent (engineered)	in vitro, synthetic assembly and L-form rebooting	[62]
ΦV10	*E. coli*	bioluminescence	Nluc	food, beef	detection of *E. coli* O157:H7 from ground beef	5 CFU/40 mL	9 h	temperate	Recombineering	[102]
K1E	*E. coli*	bioluminescence	Nluc	medical, blood	detection of *E. coli* in blood after acoustic separation	5 CFU ^†^	n/d	virulent	in vitro, synthetic assembly and reactivation in *E. coli*	[103]
NRGp5 (T7)	*E. coli*	bioluminescence	NLuc-CBM	water	detection of *E. coli*	20 CFU/100 mL	5 h	virulent	blunt cloning using T7 Select	[104]
NRGp6 (T7)	*E. coli*	bioluminescence	NLuc-CBM2a (cellulose binding)	laboratory	detection of *E. coli*	5 × 10^2^ CFU/mL	n/d	virulent	in vitro synthetic assembly and reactivation in *E. coli*	[99]
NRGp4 (T7)	*E. coli*	bioluminescence	NLuc-CBM2a	environmental, water	detection of *E. coli* in drinking water based on binding of reporter to cellulose filter	1 CFU/100 mL	10 h	virulent	direct cloning using T7Select	[60]
T7_NLC_	*E. coli*	bioluminescence	NLuc-CBM2a	environmental, water	detection of *E. coli* in water based on binding of reporter to crystalline cellulose	<10 CFU/mL	3 h	virulent	direct cloning using T7Select	[98]
T7_NLC_ (Hinkley et al. 2018, Analyst)	*E. coli*	bioluminescence	NLuc-CBM2a	food, cheese	spatial detection of *E. coli* on cheese surface	24–55 CFU/8 g	24 h	virulent	direct cloning using T7Select	[105]
T7_ALP_	*E. coli*	colorimetry	Alkaline phosphatase (ALP)	environmental, water	detection of *E. coli* in river water by phage amplification-based lateral flow assay	100 CFU/100 mL	9 h	virulent	direct cloning using T7Select	[106]
T7_ALP*_	*E. coli*	colorimetry	ALP * (*phoA D153G/D330N*) (T3 gp1.2)	laboratory	detection of *E. coli*	1× 10^5^ CFU/mL	16 h	virulent	HR + CRISPR-Cas counter selection	[107]
NRGp2 (T7)	*E. coli*	colorimetry	ALP-CBM2a: (CBM with specificity for cellulose)	environmental, water	detection of *E. coli* in water based on binding a cellulose filter	1 CFU/100 mL	10 h	virulent	direct cloning using T7Select	[60]
NRGp2 (T7)	*E. coli*	colorimetry	ALP-Cex: (Cex, exoglucanase CBM with specificity for cellulose)	environmental, water	detection of *E. coli* in water based on binding of reporter to magnetic cellulose	10^3^ CFU/100 mL	8 h	virulent	direct cloning using T7Select	[97]
T7_LacZ_	*E. coli*	colorimetry	β-galactosidase (*LacZ* operon)	food, environmental, water	detection of *E. coli* in milk, orange juice and water	10^2^ CFU/mL	8 h	virulent	direct cloning using T7Select	[64]
T7_LacZ_	*E. coli*	colorimetry	β-galactosidase (*LacZ* operon)	laboratory	detection and antibiotic resistance profiling of *E. coli*	10 CFU/mL	7 h	virulent	direct cloning using T7Select	[63]
PP01ccp	*E. coli*	colorimetry	Cytochrome *c* peroxidase (CCP)	food, various	detection of *E. coli* in lettuce, mustard greens, coriander, soybean sprouts	2 CFU/g	16.5 h	virulent	homologous recombination	[108]
IP008BK and IP052BK	*E. coli*	colorimetry	CCP	food, various	detection of *E. coli* in lettuce, mustard greens	10^2^ CFU/25 g (=4 CFU/g)	16.5 h	n/a	homologous recombination	[89]
T7_LacZ_	*E. coli*	electrochemistry	β-galactosidase (*LacZ* operon)	environmental, water, food,	detection of *E. coli* in water, apple juice and skim milk based on β-galactosidase hydrolysis of PAPG to PAP	10^2^ CFU/mL	7 h	virulent	direct cloning using T7Select	[109]
NRGp7 (T7)	*E. coli*	electrochemistry	ALP-GBP: gold-binding peptide fusion	environmental, water	detection of *E. coli* in drinking water	1 CFU/100 mL	12 h	virulent	direct cloning using T7Select	[110]
mCherrybombφ (TM4-derived)	*Mycobacterium* spp.	fluorescence	mCherry_bomb_ RFP	medical, sputum	detection of *M. tuberculosis* and phenotypic rifampicin resistance in sputum samples	20 CFU ^†^	3–5 days (126 h)	temperate, temperature sensitive	phasmids	[111]
mCherrybombφ (TM4-derived)	*Mycobacterium* spp.	fluorescence	mCherry_bomb_ RFP	laboratory	activity testing of anti-tuberculosis drugs	n/a	n/a	temperate, temperature sensitive	phasmids	[112]
Φ^2^GFP10 (TM4-derived)	*Mycobacterium* spp.	fluorescence	mVenus GFP	medical, sputum	detection of low-frequency drug-resistant subpopulations of *M. tuberculosis*	1 CFU/100,000 CFU	2 days	temperate, temperature sensitive	phasmids	[94]
Φ^2^GFP10 (TM4-derived)	*Mycobacterium* spp.	fluorescence	mVenus GFP	laboratory	drug susceptibility testing of clinical *M. tuberculosis* isolates	n/a	2–3 days	temperate, temperature sensitive	phasmids	[93]
Φ^2^GFP12 (DS6A-derived)	*M. tuberculosis*	fluorescence	mVenus GFP	laboratory	detection of *M. tuberculosis*	n/a	n/a	virulent (engineered)	phasmids	[58]
Φ^2^DRMs (TM4 derived)	*Mycobacterium* spp.	fluorescence	mVenus GFP, tdTomato RFP	medical, sputum	detection and quantitation of persister *M. tuberculosis* cells	n/d	12 h	temperate	phasmids	[92]
T7_TEV_	*E. coli*	fluorescence	Tobacco etch virus (TEV) protease	laboratory	detection of *E. coli*	10^4^ CFU/mL	3.5 h	virulent	direct cloning using T7Select	[113]
T7_ALP_	*E. coli*	fluorescence	ALP	food, various	coconut water or apple juice by fluorescent precipitated substrate for ALP coupled fluorescence imaging	100 CFU/g	6 h	virulent	direct cloning using T7Select	[114]
T7_MBP_	*E. coli*	fluorescence	Maltose-binding protein (MBP)	laboratory	detection of *E. coli* by phage amplification-based LFA	10^3^ CFU/mL	7 h	virulent	direct cloning using T7Select	[106]
PP01-TC	*E. coli*	fluorescence	Tetracysteine tag	food, apple juice	detection in artificially contaminated apple juice by fluorescent labelling of tetracysteine tag fused to capsids of progeny phage	1 CFU/mL	3 h	virulent	HR	[115]

n/d, not determined; n/a, not applicable, DETL detection limit, DETT; overall detection time; **^†^**, reaction volume not stated; CFU per well of 96-well plate; HR, homologous recombination; CBM, carbohydrate-binding module; ALP, alkaline phosphatase; CCP, cytochrome *c* peroxidase; * DETL and DETT are reported for individual assays as they were derived or provided from the source articles.

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
