# Peer review of "Reporter Phage-Based Detection of Bacterial Pathogens: Design Guidelines and Recent Developments"

_viruses, 2020, doi:10.3390/v12090944_

Round 1
Reviewer 1 Report
This is a solid effort in putting together a review on the advances of phage-based detection. While no review can be all-encompassing, this review is broad and very informative for researchers entering the field. I believe it can be published as is.
Author Response
Dear editors and reviewers,
Thank you very much for the constructive comments and suggestions for our review article “Reporter phage-based detection of bacterial pathogens: Design guidelines and recent developments (viruses-878553)”. Please find below our point-by-point response to the reviewers’ comments/questions. All changes made to our resubmitted manuscript have been introduced using tracked changes and changed to red font.
Reviewer one:
This is a solid effort in putting together a review on the advances of phage-based detection. While no review can be all-encompassing, this review is broad and very informative for researchers entering the field. I believe it can be published as is.
Many thanks to the reviewer for their kind comments.
Reviewer 2 Report
In the manuscript entitled “ Reporter phage based detection of bacterial pathogens: Design guidelines and recent developments” by Meile et. al., the authors did a very good job in exploring the most recent advancement in developing reporter-phage based biosensors for rapid, sensitive and specific detection bacterial pathogens. The article is well-written. The authors developed very nice figures to summarize different detection approaches. The literature that is reviewed is relevant to the community and sufficiently different from older reviews that it merits publication with the changes recommended below. There is a need for a review of methods of engineering reporter phages and the authors are particularly well-suited for the task.
I have the following comments and suggestions that the authors may find useful for the final published version:
- Line 76-77: I would re-write this sentence to just highlight detection of dead cells. Overestimating numbers might reflect that quantitative detection is needed in all circumstances. Presence /absence detection (of viable cells) will suffice is most cases.
- Line 81: Do you guys think that the cost element can be added here?. I believe that low profit margin in food industry is partially determining lots of their approaches to enhance food safety
- Line 93: Although true for some cases, this might be an advantage for whole active lytic phages as you guys stated later when the detection assay is based on amplifying phage signal and detecting progeny phages and/or detecting cellular enzymes .. I found that having this sentence here might lead to a confusion for the reader
- Line 114 (Figure 1) ..Line 127-128: any reason why the cell content approach wasn’t added to the figure?
- Line 147, you mention plaque assays and ELISA based assays but what about detection of phage nucleic acid after infection? Nucleic acid-based assays should also be mentioned because this sentence is summarizing the way in which increase in phage titre can be detected after infection
- Line 166-170, “… drawback poses a significant limitation …”. I would disagree with this statement. You just must pick the correct phage … phage K, Felix01, P100, etc. Selection of the phage that will be used in this type of assay and understand its interaction with its host (including BIM development) are crucial before designing and developing the assay.
Every method has its limitations and I would not say that phages that can infect >95% of strains of a species are significantly limited regarding their application in phage amplification assays, or other assays that require productive infection.
- Line -174. Please add a column highlighting the whether pre-enrichment was needed or not and for how long.
As a side note, the limitation of table 1 is that comparisons are being made between methods that used different durations of pre-enrichment. This limitation is not addressed in the text. A suitable comparison would be between methods that target a single species (or different species that have similar growth rates) and that use the same pre-enrichment duration. As it is now, the table does not allow easy comparison between different methods with respect to their detection limit. It would be preferable to have fewer examples but ones that allow more direct comparison of different detection assays in terms of detection limit.
- Line 188: What about number of the target bacteria in the samples and whether pre-enrichment is required or not? Pre-enrichment limitation hasn’t been addressed in the article. Food samples are supposed to have very few cells (if GMP has been properly followed) especially when targeting zero tolerance pathogens while samples from an infected human or animal usually have higher numbers of the infectious agent
- Line-225: Table 2: I would add the pre-enrichment column as well here ..
A511-nluc is reported to have a detection limit of 1CFU/25 g with a 24h detection time. Nucleic acid-based approaches can achieve similar sensitivities with overnight incubation. See comment about limitation of table 1.
Also, A006::nluc is reported to have 1 CFU detection limit with a three hour detection time, but the volume/weight is not given (i.e. 1CFU per what?).
- Line 228 : I believe it would be helpful and useful to the reader to provide your expert recommendations on phage genome size limits for each approach in this section ..
- Line 332: What’s “small”?
- Line 388: states that electrochemical biosensors are highly sensitive. 402-403 states that limit of detection was 105CFU/ml in drinking water after 2h. Yet in other parts of the text you mention approaches that have orders of magnitude greater sensitivity with comparable times to detection and more complex matrices (e.g. apple juice versus water). Given this, it is hard to see then how you could justify calling electrochemical biosensor-based methods highly sensitive. Please either find an example of a biosensor with a detection limit that merits the method being called “highly sensitive” or remove the characterization of electrochemical biosensors as highly sensitive.
- Any comments on reporting many examples for modified T7 reporter phages in these sections? easier system ? well established kits??
- Line 428: I am not really sure if phage resistance development during the detection assay is an major issue here..unless you mean naturally existing strains in the detection environment. In that case, it is mainly a “restricted host range “ issue that you mentioned and there is no point to specify “phage resistance” as one of the major limitation…which one do you mean?? Just please make it more clear.
- Line 41, remove “for”
- Line 62, missing reference
- Line 143, “(ie, burst size)”. Do environmental conditions not also influence e.g. the latent period? Recommend changing part in parentheses to “ie, burst size and latent period” or “e.g. burst size”
- Line 147, unnecessary coma after [51,66,67]
Line 218: recommend removing “side note” and keeping the paragraph
Author Response
Reviewer two:
In the manuscript entitled “ Reporter phage based detection of bacterial pathogens: Design guidelines and recent developments” by Meile et. al., the authors did a very good job in exploring the most recent advancement in developing reporter-phage based biosensors for rapid, sensitive and specific detection bacterial pathogens. The article is well-written. The authors developed very nice figures to summarize different detection approaches. The literature that is reviewed is relevant to the community and sufficiently different from older reviews that it merits publication with the changes recommended below. There is a need for a review of methods of engineering reporter phages and the authors are particularly well-suited for the task.
Again, many thanks to the reviewer for their kind comments.
I have the following comments and suggestions that the authors may find useful for the final published version:
- Line 76-77: I would re-write this sentence to just highlight detection of dead cells. Overestimating numbers might reflect that quantitative detection is needed in all circumstances. Presence /absence detection (of viable cells) will suffice is most cases.
We agree with the reviewer and have modified the text as such (line 76-80). “Furthermore, nucleic acid-based methods lack the ability to differentiate between DNA from viable and dead bacterial cells, meaning they can assess microbial viability retrospectively, i.e., changes in nucleic acid levels over a given timeframe; however, they are incapable of determining viability within discrete samples, making identification of only viable (and potentially infectious) bacteria extremely challenging.”
- Line 81: Do you guys think that the cost element can be added here?. I believe that low profit margin in food industry is partially determining lots of their approaches to enhance food safety
We agree that the costs associated with different bacterial diagnostics is an extremely important element for the implementation of these assays by the food industry and in the clinic; however, we believe this to be outside the scope of our review article, which aims to focus on the technical developments guiding reporter phage based assays and not their commercial implementation. The costs associated with bacterial diagnostics is also outside our area of expertise and understanding, and most importantly we do not want to provide any incorrect information to the reader here.
Line 93: Although true for some cases, this might be an advantage for whole active lytic phages as you guys stated later when the detection assay is based on amplifying phage signal and detecting progeny phages and/or detecting cellular enzymes .. I found that having this sentence here might lead to a confusion for the reader
We agree with the reviewer that our explanation about phage as “bio-probes” is misleading within this section. Our intention was to draw attention to another application of phages (and their encoded proteins) as affinity molecules for physical interaction with target cells. As such, we have changed this sentence to read (changes underlined; line 94): “The ability of phage particles to bind with high affinity and specificity to their target bacteria has led to their use as affinity molecules instead of other bio-probes such as antibodies in biosensor-based assays”.
We also modified the next sentence (changes underlined; line 96): “A drawback of using whole phage particles exclusively as affinity molecules is their relatively large size and (unless they are inactivated) their basal lytic activity that may destroy target bacteria before downstream detection can be completed.”
Line 114 (Figure 1) ..Line 127-128: any reason why the cell content approach wasn’t added to the figure?
Thanks for bringing this to our attention. We have since modified Figure 1, panel D to include the detection of cell content (indicated as small circles) alongside nucleic acid and phage progeny detection. The figure legend already mentioned cell content: “Released progeny phages [51,52] or bacterial cell content (e.g., ATP, DNA, RNA and bacterial proteins) provide excellent markers for downstream detection of the original bacterial host [53-55].”
Revised figure 1.
- Line 147, you mention plaque assays and ELISA based assays but what about detection of phage nucleic acid after infection? Nucleic acid-based assays should also be mentioned because this sentence is summarizing the way in which increase in phage titre can be detected after infection
We discuss nucleic acid-based detection later in this section (line 157 onwards); however, we agree that this should be made clearer to the reader earlier on. As such, we added the following (changes underline; line 153): “The change in phage titer can be determined by traditional plaque assays using suitable indicator strains [52,67,68], by physical detection of the phage particles using ELISA-based assays [69], or as discussed below, by measuring the increase in phage nucleic acid content.”
- Line 166-170, “… drawback poses a significant limitation …”. I would disagree with this statement. You just must pick the correct phage … phage K, Felix01, P100, etc. Selection of the phage that will be used in this type of assay and understand its interaction with its host (including BIM development) are crucial before designing and developing the assay.
Every method has its limitations and I would not say that phages that can infect >95% of strains of a species are significantly limited regarding their application in phage amplification assays, or other assays that require productive infection.
Absolutely. Finding the ideal phage with a broad host range and ability to lyse many different strains is critically important for any phage-based diagnostic. At the end of this section, we simply wanted to state that compared to reporter phage-based assays (which is the focus of our review article), the vast majority of phage amplification-based methods have the disadvantage of requiring host cell lysis for detection purposes. As such, we have changed this final statement to read (changes underlined; line 175): “A downside of most phage amplification-based assays is the requirement for completion of the phage lytic cycle to allow the release of biomarkers or progeny phages for detection. In contrast, many reporter phage-based systems do not require cell lysis or completion of the phage infection cycle to detect infected cells. Host adsorption and phage genome injection followed by reporter gene expression is generally sufficient to produce a detectable signal, allowing reporter phages to circumvent the gamut of intracellular phage defense mechanisms that can inhibit various stages of the phage infection cycle [83,84].”
- Line -174. Please add a column highlighting the whether pre-enrichment was needed or not and for how long.
As a side note, the limitation of table 1 is that comparisons are being made between methods that used different durations of pre-enrichment. This limitation is not addressed in the text. A suitable comparison would be between methods that target a single species (or different species that have similar growth rates) and that use the same pre-enrichment duration. As it is now, the table does not allow easy comparison between different methods with respect to their detection limit. It would be preferable to have fewer examples but ones that allow more direct comparison of different detection assays in terms of detection limit.
We have found numerous inconsistencies in the description of important details governing pre-enrichments, which makes side-by-side comparisons of pre-enrichment times, overall detection times, and subsequent detection specificity an extremely difficult task for different articles detailing new and varied bacterial detection methods. Importantly, this is not just limited to reporter phage-based assays. What we report in Tables 1 and 2 is the total assay time that was derived or provided from the source articles, which we believe is of most use to the reader. To make the differences between reported detection times and sensitivities of different assays more obvious to the reader, we have added the following to the footnotes of Tables 1 and 2: “*DETL and DETT are reported for individual assays as they were derived or provided from the source articles.”
- Line 188: What about number of the target bacteria in the samples and whether pre-enrichment is required or not? Food samples are supposed to have very few cells (if GMP has been properly followed) especially when targeting zero tolerance pathogens while samples from an infected human or animal usually have higher numbers of the infectious agent
Thank you for the suggestion. We have added a statement regarding the requirement of pre-enrichment for detection from samples that are expected to contain low or no microbial content. As such, we have modified point 2 of our design principles to read (changes underline; line 197):
- “Both the reporter gene and the method of detection have to be suitable for the target bacteria, expected microbial content, and the sampling environment. Inclusion of a pre-enrichment step is essential when low bacterial numbers are expected in the sample, for instance, the zero tolerance policy for common food pathogens (e.g., Salmonella, Campylobacter and coli) necessitates a pre-enrichment of food samples to ensure bacterial amplification to a minimal threshold for detection. In addition, as food components can affect fluorescence-based detection or physically block biosensors, detection from complex matrices should be coupled with an initial capture and enrichment step, e.g., bead-based magnetic separation.”
- Line-225: Table 2: I would add the pre-enrichment column as well here.
See response for comment #7 above.
A511-nluc is reported to have a detection limit of 1CFU/25 g with a 24h detection time. Nucleic acid-based approaches can achieve similar sensitivities with overnight incubation. See comment about limitation of table 1.
Also, A006::nluc is reported to have 1 CFU detection limit with a three hour detection time, but the volume/weight is not given (i.e. 1CFU per what?).
We have corrected the units to CFU/ml.
- Line 228 : I believe it would be helpful and useful to the reader to provide your expert recommendations on phage genome size limits for each approach in this section ..
Thank you for the suggestion. We have added the following:
- To section 4.1. Direct cloning (line 253): “A major limitation of using this system is the relatively small packaging capacity of the lambda phage capsid (~53 kb); meaning larger genomes are unsuitable for phasmid construction, especially for simultaneous insertion of reporter genes [121].”
- To section 4.3. Synthetic genome rebooting (line 319): “While native phage genomes of different sizes and structures can easily be rebooted in L-forms [140], the current limitation of this approach is the assembly of large (>100 kb) genomes in vitro, using approaches such as Gibson assembly of multiple genetic fragments [141].”
Line 332: What’s “small”?
We have removed “small” from the description of luciferase here. The sentence now reads, “The luciferase should be structurally stable in different environmental conditions”, which is a more appropriate description.
- Line 388: states that electrochemical biosensors are highly sensitive. 402-403 states that limit of detection was 105CFU/ml in drinking water after 2h. Yet in other parts of the text you mention approaches that have orders of magnitude greater sensitivity with comparable times to detection and more complex matrices (e.g. apple juice versus water). Given this, it is hard to see then how you could justify calling electrochemical biosensor-based methods highly sensitive. Please either find an example of a biosensor with a detection limit that merits the method being called “highly sensitive” or remove the characterization of electrochemical biosensors as highly sensitive.
We have remove “high sensitivity”; sentence now reads (line 407): “Electrochemical biosensors are relatively simple and cost-effective while remaining sensitive and specific”.
Any comments on reporting many examples for modified T7 reporter phages in these sections? easier system ? well established kits??
Thanks for the suggestion. We have added the following statement to the end of this section to hopefully clarify the many T7-based assays having been developed recently (line 268): “Not only is a commercial kit available for T7 phage engineering (i.e., T7Select® (Novagen)), but as a model phage it has a genome suitable for insertion of heterologous genes and a well-characterized host range, making it an ideal phage for reporter phage-based assay development as can be observed by its frequent use over the last four years (Table 2).”
- Line 428: I am not really sure if phage resistance development during the detection assay is an major issue here..unless you mean naturally existing strains in the detection environment. In that case, it is mainly a “restricted host range “ issue that you mentioned and there is no point to specify “phage resistance” as one of the major limitation…which one do you mean?? Just please make it more clear.
We agree with the reviewer that phage resistance is unlikely to develop during enrichment from a food or clinical sample; however, it would always be unclear based only on phage amplification if a phage resistant (and therefore undetectable) subpopulation of bacteria is also present that goes undetected. Our statement here has been changed as follows to better explain the link between host range and phage resistance: “Nevertheless, the major limitations to practical reporter phage application are food matrix effects and the restricted host ranges of many phages that is intrinsically linked to phage resistance.”
- Line 41, remove “for”
Corrected.
- Line 62, missing reference
Thanks for bringing this to our attention. The following reference has been added:
- Soon, J.M.; Brazier, A.K.; Wallace, C.A. Determining common contributory factors in food safety incidents–A review of global outbreaks and recalls 2008–2018. Trends in Food Science & Technology 2020, 97, 76-87.
Line 143, “(ie, burst size)”. Do environmental conditions not also influence e.g. the latent period? Recommend changing part in parentheses to “ie, burst size and latent period” or “e.g. burst size”
This has been corrected to include latent period (line 148): “(i.e., burst size and latent period)”
- Line 147, unnecessary coma after [51,66,67]
Section has been modified as described in our response to comment #3, which now warrants the use of a comma here.
Line 218: recommend removing “side note” and keeping the paragraph
Changed to read “In addition, various reporter proteins…”
